# Public Attitudes towards Forest Pest Damage Cost and Future Control Extent: A Case Study from Two Cities of Pakistan

**Umer Hayat** [1] , **Aqsa Abbas** [2] **and Juan Shi** [1,*]

1    Sino-France Joint Laboratory for Invasive Forest Pests in Eurasia, Beijing Forestry University,
     Beijing 100083, China; oomarcassi6116@gmail.com
2    Visiting Faculty, Faculty of Agricultural Sciences, University of The Punjab, Lahore 54590, Pakistan;
     aqsaabbas.ibit@pu.edu.pk
*    Correspondence: shi_juan@263.net

**Abstract:** Infestations of pests are perhaps an anthropogenic catastrophe for trees. *Aeolesthes sarta* (Sart longhorned beetle—SLB) is one of the most severe pests that cause serious damage to a number of hardwood tree species, i.e., *Populus*, *Salix*, *Acer*, *Juglans*, and *Malus*. To investigate people's attitudes towards pest damage cost and future control extent of SLB, a door-to-door method was adopted to survey two major cities (Quetta—QU and Peshawar—PE) of the northwestern region of Pakistan where this pest has caused severe damage. Respondents were asked about SLB pest knowledge, pest damage costs, preferences for control choices, and program extent. According to respondents, more trees (181 ± 1.20 trees/ha/annum) were damaged in QU compared to PE. *Populus* spp. was the dominant tree genre that attacked and damaged the most. Around 85% of respondents from both cities stated the pest damage cost was calculated as high for QU (480,840.80 ± 4716.94$/annum) compared to PE. Respondents in both locations strongly supported (more than 82%) biological control of future SLB outbreaks. They all agreed that protecting ecologically vulnerable places and wildlife habitats should be the primary priority in a future SLB outbreak. Respondents from both cities who preferred to protect more land area in future SLB outbreaks were calculated to be high for QU (61%) compared to PE (58%). However, city variations in opinions regarding forest-type priority that should be protected and control options were observed. Socio-demographic characteristics were discovered to impact pest damage cost positively, as well as preferred SLB control extent. The findings of this study can help policymakers and forest managers develop publicly permissible pest control plans and make more accurate predictions about future pest outbreaks.

**Keywords:** *Aeolesthes sarta*; *Trirachys sarta*; Sart longhorned beetle (SLB); quetta borer; Quetta; Peshawar; questionnaire; survey; pest damage cost; future pest control extent





## 1. Introduction

Infestations of pests are perhaps an anthropogenic catastrophe for trees. The most often implied pest species correspond to the Lepidoptera and Coleoptera orders [1]. The Cerambycidae family of the Coleoptera order, which contains longhorn beetles, is one of the world's most broadly diversified, ecologically and economically notable pest families [2]. *Aeolesthes sarta* or *Trirachys sarta*, often known as the Sart longhorned beetle (SLB), is one of the predominant species of the Cerambycidae family [3–5]. It is a polyphagous pest that mainly preys on broadleaved tree species belonging to the genus *Populus*, *Juglans*, *Acer*, *Salix*, *Malus*, *Platanus*, and *Ulmus* [3]. This family's larvae are internal feeders, mainly feeding on the plants' living or dead tissue [6]. Larval boring tends to break down host trees structurally and obstruct the flow of nutrients and water, which causes killing of multiple branches and, in severe cases, destruction of a whole tree [7–10].

Before swiftly expanding into Afghanistan, Iran, and other Central Asian countries, it is speculated that SLB originated in Pakistan and the western part of India [11,12].

Warm climates and the particular types of host trees the invasive pests favor promote their establishment [13]. This species might prove highly dangerous in areas with hot, dry weather [14]. Infestations are especially noticeable in mountain forests, which may be a factor in the decline of poplar tree forests, an important wood supplier for the market and industry for wood [15]. Within two to three years, a substantial infestation can lead the affected trees' canopy to decline and the foliage to dry out [16].

SLB feeds on 15 distinct kinds of trees, making it one of India's most significant pests of hardwood tree species in both natural and artificial forest stands [17]. This beetle is one of the foremost destructive pests of walnut trees (*Juglans regia*) in India [18]. It is a serious economic problem in Iran and the Kashmir region [19]. In Turkmenistan, apple orchards (*Malus* spp.) and shelter belts have also sustained substantial economic loss due to SLB attacks [20,21]. In the Beshkent and Vakhsh Valleys of Tajikistan, it is difficult to find trees that are not impacted by SLB [22]. At higher elevations (>1800 m.a.s.l.), the rate of infestation is observed to be decreased, while isolated infected trees have been found frequently [22]. SLB imposes the greatest destruction in urban areas, where trees are less resistant to pests and thrive in challenging environmental conditions (poor drainage, close distance to a road, etc.) [3]. In fact, in Tashauz (Turkmenistan), SLB killed a substantial number of tall trees at urban sites [23].

In Pakistan, just like many other broadleaved tree species, i.e., *Populus*, *Salix*, *Acer*, *Platanus*, etc., shelter belts and apple orchards have also been significantly infested by SLB [3,24]. In western regions of Pakistan, this pest was found in apple orchards for egg laying [24]. In the region of the west of Pakistan, between 1900 and 1907, SLB destroyed and killed over 3000 trees of poplar, willow, and elm, and the beetle became renowned in Quetta and throughout Baluchistan (where it is known by the moniker "borer") [3,24,25]. SLB is one of Pakistan's deadly poplar borers [26] and has seriously harmed a significant number of Populus plantations [15,26,27].

This study set out to perform an extensive, multiregional investigation of public perceptions of pest outbreaks, damages, and management strategies. In 2022, we surveyed the public in Quetta (QU) and Peshawar (PE) to understand and compare the damage cost and attitudes regarding the control of a pest species [*Aeolesthes sarta*—Sart longhorned beetle (SLB)] that has caused serious damage in both cities. This pest and these regions were selected for numerous reasons: (1) this pest is one of the lethal forest pests of hardwood stands in Pakistan; (2) SLB outbreaks cause significant branch kill and often result in tree death [15,25–27], and thus, SLB outbreaks typically have a considerable impact on communities that depend on forests, the forest sector, and the supply of timber; and (3) populations in QU and PE have notably diverse experiences with this pest since QU has had more severe SLB outbreaks than PE has.

Several key research questions were explicitly addressed in this study: (i) To what extent is the general population aware of and informed about forest pests (e.g., aware of previous pest outbreaks, aware of SLB)? (ii) What are public perceptions toward SLB outbreak damage (e.g., preferred forest stand types to be attacked, preferred tree species to be attacked, average number of trees to be damaged/ha, and approximate damage cost due to SLB attack)? (iii) How do people generally feel about controlling the SLB outbreak (e.g., type of forest preferred to conserve, preferred means of control, desire to support protection, desired protection extent)? (iv) Do public opinions on SLB control strategies vary by region? (v) What elements affect the scope of the public's suggested control program for future SLB outbreaks?

We developed two models related to several socio-demographic features, forest stand, and type-related variables to examine factors influencing public perceptions over SLB damage cost and public preferences over control program extent. For an idea, we followed up the model used by Chang et al. [28] in their study (for further elaboration, please refer to the complete paper authored by Chang et al., 2009). We hypothesized the following: (1) respondents who had better knowledge about SLB (it can kill trees) and its damage, i.e., type of forest stand damaged, type of species damaged, and the number of trees

damaged, would propose a large number as damage cost caused by pest outbreak, and prefer the large area to be protected from future SLB outbreak; (2) those associated with the government forest department would prefer more forest land protected from an SLB outbreak; (3) males, younger people, and people with more experience would opt for protecting a greater percentage of forest land from future SLB outbreaks, and people with higher levels of education (in our case, forest department workers, university students, and teachers/professors were considered as highly educated people) would support doing the same; and (4) cities of residence would not have any bearing on protection area preference because there was insufficient relevant literature to base a difference on.

## 2. Materials and Methods

### 2.1. Study Area

We examined public opinion about the damage cost due to the SLB attack, attitudes about the SLB outbreak, control options in two grand cities, Quetta (QU) and Peshawar (PE) of Pakistan (Figure 1). QU is the capital city of a Baluchistan province located in the west of Pakistan, which has an area of 3501 km$^2$ and a population of 1,001,205 [29]. Its elevation ranges from 1388 to 3470 m.a.s.l., and it lies in a dry temperate zone with cold winters and moderately hot summers and mostly receives precipitation between January and April [30] (Figure 2). The vegetation cover of QU is almost 32,987 ha [31], covering around 25% of QU's total area. Out of this 25%, 17.5% of the area is natural forest owned by the provisional/federal government, and the remaining 7.5% consists of privately owned woodland and farmlands. In contrast, PE is the capital city of a Khyber Pakhtunkhwa (KPK) province located in the north-west of Pakistan, which has an area of 1257 km$^2$ and a population of 1,970,042 [29]. Its elevation ranges from 280 to 682 m.a.s.l., and it lies in the sub-tropical zone with moderately cold winters and hot summers and mostly receives precipitation in the early and middle months of the year [30] (Figure 2). The vegetation cover of PE is 46,221 ha [32], covering around 36.77% of PE's total area. Of this, 30.15% of the area is natural forest owned by the provisional/federal government, and the remaining 6.62% consists of privately owned woodland. Being capital cities of their provinces, QU and PE contribute majorly to the wood, paper, and furniture industries. Historic SLB outbreaks in QU and PE occurred in a cyclical pattern in 1906–1907, 1976, and 1980. These outbreaks affected thousands of *Populus*, *Malus*, *Salix*, *Acer*, and *Platanus* trees in natural forestland, woodlands, and farmlands, respectively [3,15,24–27]. The severity of SLB historic outbreaks was high in QU compared to PE.

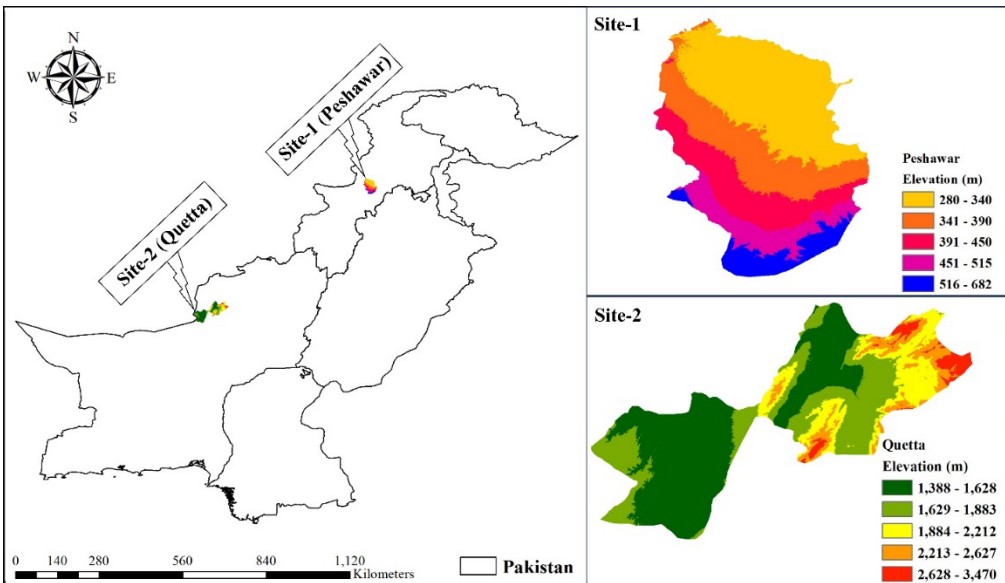

**Figure 1.** Site map of study cities of Pakistan. DEM file source: https://opentopography.org/.

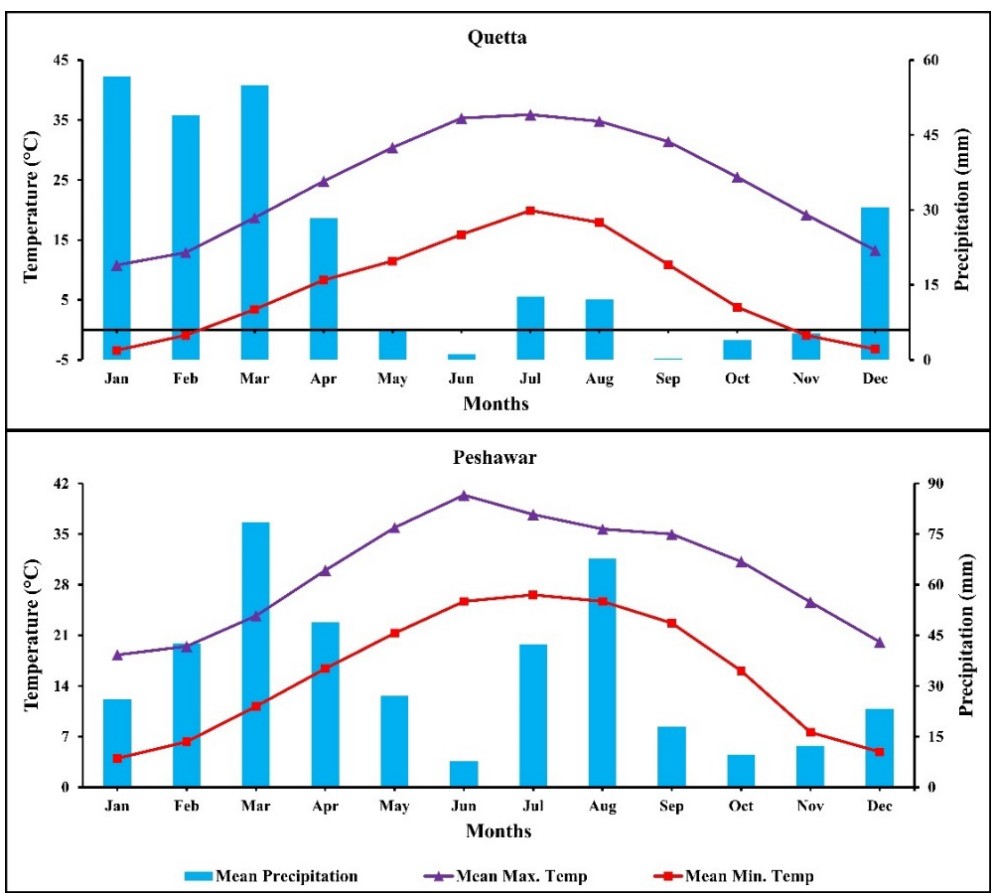

**Figure 2.** Metrological data of study cities of Pakistan. Source: https://www.pmd.gov.pk/en/ (PMD, 2022).

### 2.2. Protocols of Survey and Questionnaire Design

A door-to-door survey method (presenting a questionnaire and brief interview) was adopted to conduct this survey study [33]. To address the objectives of the study well, a selective sub-group from the large population was shortlisted from each city based on their basic knowledge about the forest pests based on sampling size calculated by the formula (Equation (1)) [34]

$$n = \frac{N}{1 + N(e)^2} \tag{1}$$

where n = sample size, N = total population (here N = 1,001,205 for QU, and N = 1,970,042 for PE), and e is the acceptable sample error (here e = 0.05). A total of 399 individuals from that sub-group were randomly sampled in each city. The questionnaire was composed of four major sections: (1) socio-demographic information of respondents; (2) questions about forest values and their importance, familiarity with pests, and experiences with pest disruptions; (3) respondents' opinions about the damage caused by SLB and personal satisfaction about pest control programs; (4) respondents' preferences for the types of forests to be protected, what kinds of controls to be used, and the scope of protection in the case of future SLB outbreaks (Table 1). The first part of the survey requested participants to provide socio-demographic data, including their gender, age, employment designation, knowledge and experience, and place of residence. The extent to which socio-demographic characteristics significantly affected perceptions concerning forest insect damage and control within and across cities was investigated using responses to the questions asked in the second, third, and fourth parts of the questionnaire. In the second segment of the survey, responses to questions about the forest values were presumed to be able to shed light on public perceptions about pest outbreaks and pest control. Researchers and entomologists

familiar with the pest assisted in developing the knowledge assessment questions based on prior contents. The third section of the survey was accompanied by a brief overview of damage caused by previous outbreaks in the respondent's city based on the literature review [3,15,24–27]. With this information, respondents were asked to choose one most preferred type of tree species genera suitable for an SLB attack: *Populus*, *Salix*, *Juglans*, *Acer*, and *Malus*. Based on the literature, respondents were addressed about the severity of damage and asked to choose one most preferred option about the number of trees/ha that have been damaged due to the SLB attack annually: <50 trees, <100 trees, <150 trees, and >150 trees. Respondents were presented with the mathematical calculation of possible damage cost based on the number of trees damaged/ha and the total area that has been attacked; based on that, respondents were asked to choose one out of four most preferred values as damage cost due to SLB attack in their city: <100,000$, <250,000$, <500,000$, and >500,000$. Respondents were asked to rate their satisfaction with the federal/provisional SLB control program based on five scale points. Respondents were presented with four options of the federal/provisional funding amount in controlling SLB and asked to choose one most preferred digit as the funding amount: <10,000$, <50,000$, <100,000$, and >100,000$. To eliminate information bias, the fourth section of the survey was accompanied by a brief overview of SLB attributes and factual data regarding previous outbreaks and vulnerable forest lands in the respondent's city. With this knowledge, respondents were asked to select their top choice of type of forest for prioritization of protection during an outbreak: productive forests; protective/reserved forests; ecologically sensitive and wildlife habitat areas; and recreation areas. Brief explanations of the attributes of the control option were presented before the questions on preferences in management selections.

**Table 1.** Description of the survey's inquiries.

| Questions | Scale/Units/Categories |
|---|---|
| **Socio-demographic Statistics** | |
| Gender, Age, Designation, Experience, Location | Categories [a] |
| **Forest value** | |
| (1) Provide timber and firewood; (2) provide jobs; (3) help maintain biodiversity; (4) fulfill the local community's demand. | 4 scale (1 = not important, 4 = very important) |
| **Experience and Knowledge of SLB pest** | |
| Did you ever visit any forest land impacted by forest pests? | Yes/no |
| Did you ever visit any forest land affected by SLB? | Yes/no |
| Are you aware of pest outbreaks in (your city) forests before? [b] | Yes/no |
| Are you aware of any before SLB outbreaks in (your city) forests? [b] | Yes/no |
| Did you know SLB is a boring pest that creates a tunnel inside the tree and kill it? | Yes/no |
| Did you know SLB attacks and damage hardwood tree species? | Yes/no |
| Did you know SLB is native to Central Asia? | Yes/no |
| **Information about damage caused by SLB** | |
| What type of stand is suitable for a SLB attack in your city? | 1 = pure stand, 2 = mixed stand, 3 = both |
| Which broadleaved tree species mainly attacked in your city? [c] | 1 = *Populus*, 2 = *Salix*, 3 = *Juglans*, 4 = *Acer*, 5 = *Malus* |
| On average, how many trees/ha/years damaged by SLB? [d] | 4 categories (<50–>150trees) |
| Approximately how much does wood damage due to SLB attack annually cost? [e] | 4 categories (<100,000–>500,000$) |
| **Pest control program** | |
| Are you satisfied with the government pest control program for SLB? | Yes/no |
| How much did the government fund for the SLB control program? [f] | 4 categories (<10,000–>100,000$) |
| **People's attitude toward SLB control** | |
| Do you support government financing for research and development on SLB control? | Yes/no |
| Will you support controlling future SLB outbreaks in your state? | |
| What percentage of the forestland should be protected if an SLB outbreak emerges in a forest area in your city? | 5 categories (15%–75%) |

**Table 1.** *Cont.*

| Questions | Scale/Units/Categories |
|---|---|
| **Management options and respondents' preferences** | |
| Which control measure for SLB outbreaks would you wish to see implemented? | 1 = synthetic chemical control, 2 = biological insecticide, 3 = biological control agent, 4 = silvicultural treatments |
| Which type of forest should be conserved against SLB outbreaks first if funds are limited? | 1 = productive forests, 2 = protective/reserved forests, 3 = ecological sensitive and wildlife habitat areas, 4 = recreational forests |
| **Attitude on who should cover the cost of a program for controlling pests** | |
| Who should pay the cost of the SLB control program on forestland and privately owned land? [g] | 1 = government, 2 = land owner + private organizations, 3 = both |
| How often should a survey of forestland at a local or provisional level be conducted to monitor SLB attacks in a timely manner? | 1 = once in a year, 2 = once in two years, 3 = once in three years, 4 = once in five years |

[a] Categories, gender = 2 (male/female); age = 4 (<25 to >45 years old); designation = 5 (farm owner/farm worker/government forest department/university student/university teacher); experience = 4 (<5 to >15 years), and location = 2 (Quetta/Peshawar). [b] Those who responded "Yes" were asked to list the specific pest outbreaks they were aware of. [c] Tree species were selected based on the literature review [3,15,25–27]. [d] Figure for the average number of damaged trees was selected based on the literature review [3,15,26,27]. [e] Approximate figures for the cost of damage were selected using the actual price value of a single healthy log of *Populus, Salix, Juglans, Acer,* and *Malus* tree species, e.g., one mature, healthy log of *Populus* tree species is about 35–50$. [f] Approximate figures for the fund program cost were selected after detailed discussions with several federal and provincial forest department officials. [g] Respondents had the choice to say if they preferred not to pay.

Respondents were explicitly educated that biological insecticides such as *Bacillus thuringiensis* (B.t.), which has been utilized in many areas to manage pest outbreaks and is not dangerous to humans, have been used to control pest outbreaks [28]. Synthetic chemical control such as oxydemeton-methyl (organophosphates) has been proven very useful for controlling SLB [8], but excessive use of chemicals could be dangerous for tree and human health, and every insect that comes into acquaintance with it dies [3]. A biological control agent, or *Beauveria bassiana*, effectively controls SLB adults [14,35]. Silvicultural treatments entail growing trees of highly resilient species and cutting down and burning infected trees [3,14,36]. After being briefed, those surveyed were asked to specify their preferred option for SLB control. Before usage, the reliability and validity of the questionnaire were evaluated in two focus group sessions held in April 2022 (one for pest professional sessions and one for selected sub-group members). However, the pilot survey study that resulted in the Cronbach alpha value of "a = 0.80 (n = 50)" demonstrated the validity and reliability of the questionnaire. The survey was carried out in two stages, the first in May 2022 and the second in June 2022. Seven hundred ninety-eight questionnaires were used and filled during the survey (399 per city). To mitigate the possibility of bias when presenting findings, the sample attribute was factored in cases where excess or underrepresentation of a key population feature occurred in the sample [37].

*2.3. Data Analysis*

In descriptive statistics, means and S.E (standard error) by cities were determined for responses to forest values, funds for the SLB control program, average number of trees damaged/ha, damage cost, and protected area preference, frequencies and S.E by cities were calculated for the type of stand damage, type of species attacked/damage, preferred control methods, type of forest to be protected, attitude towards damage cost payment, and number of field survey annually. Mann–Whitney–Wilcoxon U *t*-test was used to investigate variations in city means. The correlation between the city of residence and a number of categorical factors, such as pest outbreak experiences, pest knowledge, type of stand damage, type of tree species damage, satisfaction with a control program, support for government funding research and development to enhance control measures, support for controlling future pest outbreaks, the duty to pay pest control cost, and forest-type protection, were investigated using the chi-square test of homogenization (see Table 1

for categories). IBM SPSS Statistics (Version 26) * (* IBM Corp. Released 2019. IBM SPSS Statistics for Windows, Version 26.0. Armonk, NY, USA: IBM Corp) was used for all analyses.

There were two multivariate regression models used in this study [28], one to examine factors that influence public opinion towards forest damage cost due to SLB attacks in each city and the other to investigate variables that affect public preferences on the scope of a control program in the event of SLB outbreaks in the future in each city. For damage cost analysis, the dependent factors cast in the regression analysis were the approximate currency figures used in the questionnaire (for detail, please see Section 2.2 and Table 1) (from <100,000$ to >500,000$) that have been cost due to the SLB damage annually. The factor was <100,000$ for those who indicated no or slight damage cost due to pest outbreaks and ranged from 100,000 to >500,000$ for those who indicated significant damage cost due to pest outbreaks.

For future control program extent analysis, a dependent factor castoff in regression analysis was the preferred land % of forest (from 0 to 75%) that should be shielded from future SLB outbreaks. The factor was set at <15% for those who showed no or significantly less interest in future pest outbreaks and extended from 15 to 75% for those who favored pest outbreak control in the future and stated their preferred future land % of forest that should be guarded.

For pest damage cost analysis, independent variables were stated as follows: (i) a 'forest values' ordinal variable, calculated by summing all the scores of forest uses and values; (ii) knowledge of the pest dummy variable was determined using a participant's accurate or inaccurate affirmation of a statement on SLB attributes (see Table 1); (iii) an array of dummy variables were used to quantify demographic information for gender (1 = male, and 0 = female), age (1 = >45 years old, and 0 = <45), designation (1 = work in government forest department, and 0 = farm owner, worker, student, professor/teacher), experience (1 = >15 years and 0 = <15 years), location (1 = PE, and 0 = QU); (iv) satisfaction towards funding program, measured as dummy variable (1 = yes and 0 = no); (v) government pest control funding program amount, measured as dummy variable (1 = >100,000$ and 0 = <100,000$); (vi) type of stand damage, measured as dummy variable (1 = Pure stand, 0 = mixed, both or none of these); (vii) type of tree species damaged, measured as dummy variable (1 = *Populus* spp. and 0 = *Salix*, *Juglans*, *Acer*, and *Malus* spp.); (viii) number of tree damage/ha, measured as dummy variable (1 = >150 trees and 0 = <150 trees). While for future control program extent analysis, (i, ii, iii) were used as the same, other independent variables such as (iv) support for SLB future outbreak program, measured as a dummy variable (1 = yes and 0 = no); (v) preferred control method, measured as a dummy variable (1 = biological agent and 0 = synthetic chemical control, biological insecticide, and silvicultural treatments); (vi) preferred forest type to be protected, measured as a dummy variable (1 = Ecologically sensitive and Wildlife habitat forests and 0 = productive forest, protective/reserved forests, and recreational forest); (vii) preference of paying control program cost, measured as a dummy variable (1 = government and 0 = landowner, both, and none of these); and (viii) preference about the forest survey duration, measured as a dummy variable (1 = once in a year and 0 = once in two years, once in three years, and once in five years), respectively. Because of the moderate associations between the independent variables ($r = 0.5$), collinearity was not a problem [28]. Levene's tests were used to look for potential heteroskedasticity in the variance of the samples' error terms, and the findings showed that the null hypothesis of homoskedasticity could not be rejected at the significance level of 0.01.

## 3. Results

### 3.1. Socio-Demographic Features

The socio-demographic features of the survey samples in QU and PE largely represented the total populations in each city (Table 2). Overall calculated sample size n = 798 (n = 399 for each city) was used to conduct the survey and collect data for analysis. Accord-

ing to the chi-square test, all the socio-demographic features of both cities concerning each other were measured as statistically significant ($p < 0.01$). Male respondents were high in both cities compared to female respondents, with a maximum frequency of 284 (71%) for QU. In both cities, approximately 65% of respondents were from the age group between 25 and 45 years; however, the number of respondents from the age group > 45 years accounted for 98 (25%) for QU compared to PE. A considerably more significant percentage of respondents from both cities were students (29% in QU and 36% in PE), while the number of respondents working as professors/teachers was higher in PE, 27%, compared to QU. Additionally, significantly more respondents from PE (30%) had experience > 15 years compared to QU (27%).

**Table 2.** Socio-demographic features of Quetta and Peshawar survey sample.

| Socio-Demographic Features | Quetta | | Peshawar | |
|---|---|---|---|---|
| | **Frequency** | **% [a]** | **Frequency** | **% [a]** |
| **Gender [b]** | | | | |
| Male | 284 | 71 | 241 | 60 |
| Female | 115 | 29 | 158 | 40 |
| **Age [b]** | | | | |
| <25 y | 40 | 10 | 75 | 19 |
| 25–35 y | 133 | 33 | 128 | 32 |
| 36–45 y | 128 | 32 | 135 | 34 |
| >45 y | 98 | 25 | 61 | 15 |
| **Designation [b]** | | | | |
| Farm owner | 53 | 13 | 25 | 6 |
| Farm worker | 81 | 20 | 54 | 13 |
| Govt. Forest Department | 76 | 19 | 72 | 18 |
| University Student | 112 | 29 | 144 | 36 |
| University Teacher | 77 | 19 | 104 | 27 |
| **Experience [b]** | | | | |
| <5 y | 93 | 23 | 85 | 21 |
| 6–10 y | 72 | 18 | 113 | 28 |
| 11–15 y | 125 | 32 | 82 | 21 |
| >15 y | 109 | 27 | 119 | 30 |

[a] Percentage of the population based on the total survey sample size of 399. [b] Symbolizes that frequency distributions and percentages are statistically different for Quetta and Peshawar at $p < 0.01$ (Chi-square test).

*3.2. Forest Values*

Respondents from both cities indicated high scores for the importance of forests (Figure 3). Those from QU indicated a significantly higher value of the forest for maintaining biodiversity than the PE respondents. Respondents from QU valued the forest significantly more for providing jobs to the local community than PE respondents. However, respondent scores from both cities about the forest value as providing timber and fuelwood and fulfilling the local community's demand were the same (no significant difference). According to the mentioned statistics on socioeconomic factors and forest values, there were significant variations between the QU and PE respondents. This paved the way for an insightful city-to-city comparison of public opinion and perceptions of forest pest outbreaks, damage cost, and control.

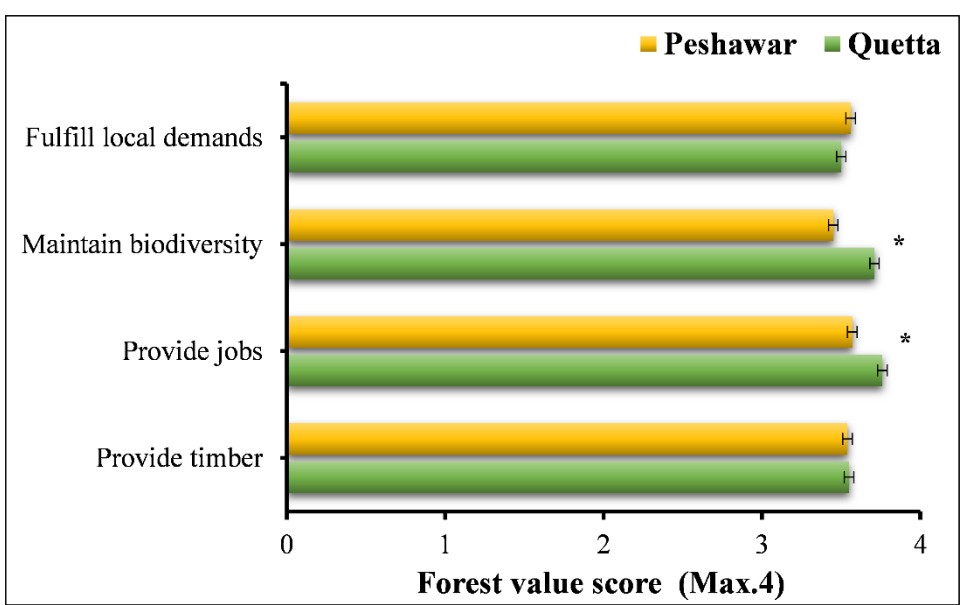

**Figure 3.** Survey respondent results interpreting forest values in Quetta and Peshawar. The bar shows the mean response value of respondents regarding forest values. The bars display ±0.03 standard errors (S.E.) on either side of the mean. * Represents that city means were statistically different at $p < 0.05$ (Mann–Whitney–Wilcoxon U-test).

### 3.3. Familiarity with SLB and Encounters with Forest Pest Disruptions

Approximately 50% of respondents from each city asserted that they had been to a forestland where pests invaded (Table 3). In contrast to PE respondents (61.3%), a significantly higher proportion of QU respondents (69.8%) asserted they were aware of prior insect outbreaks in their city. Moreover, a considerably higher proportion of QU respondents (81.9%) reported knowledge of previous SLB outbreaks in their city than PE (71.5%). Several intriguing conclusions appeared when comparing SLB knowledge across cities (Table 3). Significantly more QU respondents knew that (i) "SLB is a boring pest that creates tunnels inside trees and often kills trees" (73.8% in comparison to PE 66.1%); (ii) "SLB is the most destructive pest of hardwood tree species" (68.1% in comparison to PE 60.3%); however, responses from respondents relative to the origin of the pest from both cities were nearly identical (52.1% in comparison to PE 50.3%).

**Table 3.** Experience with disruptions caused by forest pests and familiarity with SLB in Quetta and Peshawar among survey respondents.

| Experience and Knowledge | Quetta | | Peshawar | |
|---|---|---|---|---|
| | N | % [a] | N | % [a] |
| **Experience with pest disruption [b]** | | | | |
| Have you ever been to a forest where pests have been a problem? | 376 | 51.2 | 365 | 49.7 |
| Are you informed of any pest outbreaks that have previously happened in your city? | 359 | 69.8 * | 338 | 61.3 * |
| Are you informed of any SLB outbreaks that have previously happened in your city forests? | 351 | 81.9 * | 330 | 71.5 * |
| **SLB Knowledge [b]** | | | | |
| Is a boring pest creating a tunnel inside the tree and killing it? | 375 | 73.8 * | 365 | 66.1 * |
| Attack and damage hardwood tree species? | 371 | 68.1 * | 364 | 60.3 * |
| Is a pest native to central Asia? | 311 | 52.1 | 307 | 50.3 |

* Demonstrated that frequency distributions between cities are statistically different at $p < 0.05$ (Chi-square test). [a] Respondents percentage that chose "yes" when asked a question. [b] Employed a "Yes" or "No" question structure for evaluation.

### 3.4. Respondent's Opinion about Forest Damage Cost and Personal Satisfaction towards Funding Program

Respondents from both cities stated that pure stands were attacked more frequently and more susceptible to SLB attacks (Figure 4a); however, compared to PE (10%), 34% of respondents from QU significantly stated that both 'pure and mixed' stands were attacked frequently and susceptible to SLB attacks. A total of 49% of respondents from QU compared to PE (37%) significantly highlighted that *Populus* tree species were attacked and damaged most in their city (Figure 4b); however, more than 35% of respondents from both cities asserted that *Populus* spp. was the main tree species damaged more often. While in PE, 26% of respondents identified *Malus* species as the second most damaged tree species in their city compared to QU (13%). According to respondents' perceptions, the tree damage rate due to SLB attacks was higher in both cities (Figure 4c). However, 53% of respondents from QU stated that more than 150 trees/ha were damaged annually, significantly higher than PE (46%). A total of 50% of respondents from both cities stated that government funding for the SLB control program was less than 50,000$ (Figure 4d); however, the amount for the SLB control program was calculated higher for PE than QU.

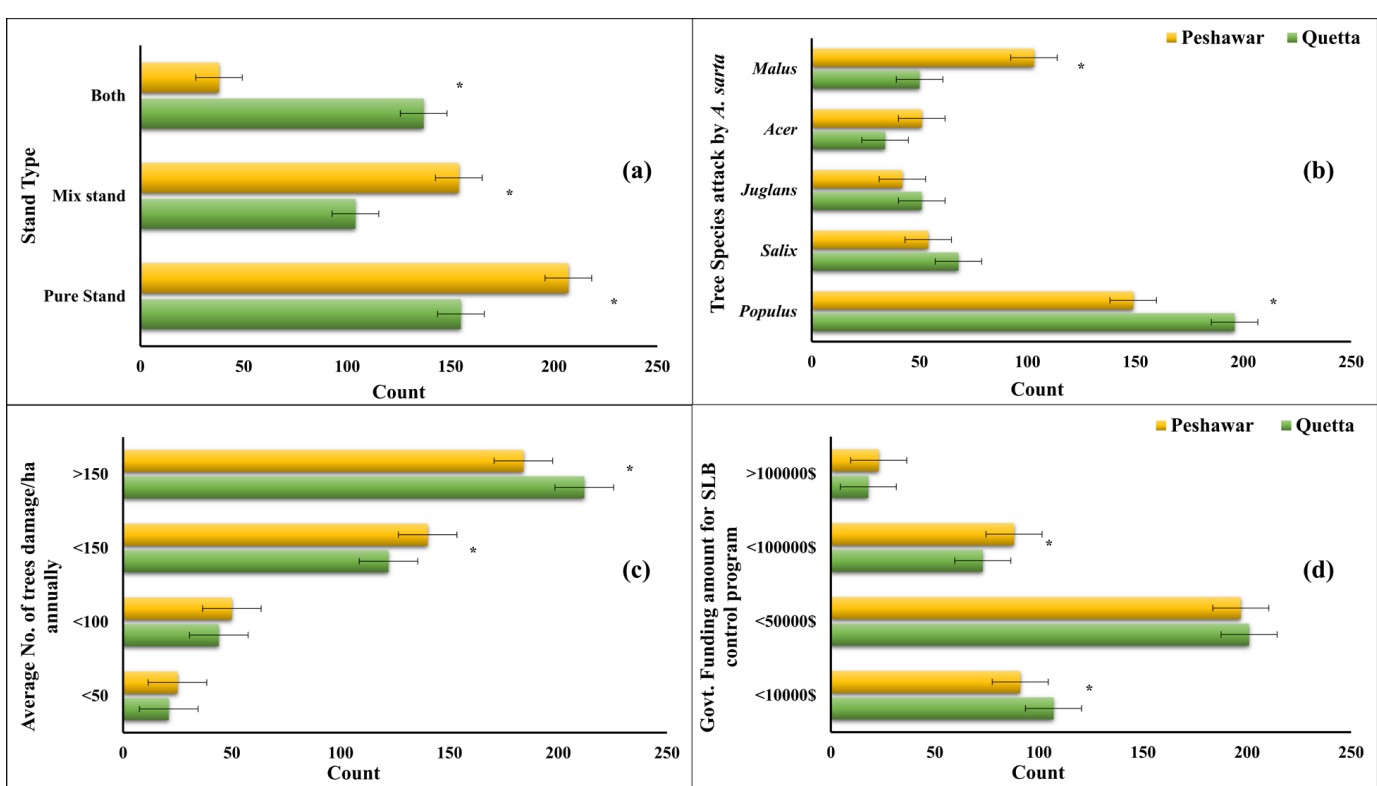

**Figure 4.** (**a**) Survey respondents' opinions regarding forest stand types that attacked the most, (**b**) type of tree species that attacked the most, (**c**) average number of trees/ha damage annually, (**d**) and government funding amount for the SLB control program in Quetta and Peshawar. Bars represent frequencies ± SE. * Indicating that frequencies differed statistically between cities at $p < 0.05$ (chi-square test).

Regarding respondent satisfaction rate towards the government SLB control program and funding, less than 50% of respondents in both cities responded in favor of the government SLB control program and funding (Table 4), especially in QU, where only 44.4% responded showed satisfaction towards the government funding and control program compared to PE (49.6%). Furthermore, based on the respondent satisfaction rate, the funding amount for the SLB control program was also reported to be significantly less in QU (49,022.55 ± 1471.21$) compared to PE (54,461.15 ± 1582.83$). According to respondent responses, the average number of trees that have been dam-

aged annually was (181 ± 1.20 trees/ha/year) in QU, which was significantly higher than PE (168 ± 1.73 trees/ha/year). The average cost, which was calculated based on respondent responses, was (480,840.80 ± 4716.94$) in QU, significantly higher than PE (453,369.19 ± 6886.86$) (Table 4).

**Table 4.** Respondents' satisfaction and response toward government funding for SLB damage control, average number of tree damage, and forest damage cost.

| Satisfaction Rate and Responses | Quetta | | Peshawar | |
|---|---|---|---|---|
| | N | Value | N | Value |
| **Personal Satisfaction:** | | | | |
| Are you satisfied with the government pest control program? [a] | 399 | 44.4 * | 399 | 49.6 * |
| What is the average funding amount for pest control in your city? [b] | 399 | 49,022.55 (1471.21) ‡ | 399 | 54,461.15 (1582.83) ‡ |
| **The response towards pest damage:** | | | | |
| On average, how many trees are damaged/ha annually? [c] | 399 | 181 (1.20) ‡ | 399 | 168 (1.73) ‡ |
| What is the approximate forest damage cost? [b] | 399 | 480,840.80 (4716.94) ‡ | 399 | 453,369.19 (6886.86) ‡ |

* Demonstrated that frequency distributions between cities c at $p < 0.05$ (Chi-square test). ‡ Demonstrated that distribution means between cities differ significantly at $p < 0.05$ (Mann-Whitney-Wilcoxon U *t*-test). [a] Percentage of respondents chose "yes" when asked a question. [b] Mean amount presented in US$ with standard error in brackets. [c] Mean number of trees damage/ha with standard error in brackets.

A multivariate regression analysis of features influencing public response over damage cost that has been caused due to SLB damage in their cities is presented in Table 5. Overall, the coefficients of the regression model were statistically significant. All the variables positively influenced public opinion about pest damage costs except designation, location, personal satisfaction towards the government SLB control program, and government funding. Those who were males, those who had 15 years of experience and more, those who had rated the forest value high, those who had pest knowledge, those who had chosen pure stand as most suitable for SLB attack, those who had chosen *Populus* spp. as the most attacked tree species by SLB, and those who had stated 150 or more trees/ha damaged by SLB stated a significantly high damage cost. However, those who worked other than the government forest department, those who were QU residents, those who were not satisfied with the government SLB control program/funding, and those who indicated a government funding amount of less than 100,000$ stated a significantly high damage cost. Age did not have any influence on respondents' opinions. The value of the F-statistic in regression (F = 130.89, $p < 0.001$) demonstrated that the null hypothesis of no association between the dependent and independent variables could be reliably ignored. The variables featured in the regression outlined 66% of the variability concerning SLB damage cost.

**Table 5.** Regression results of factors influencing damage cost of the forest due to the *Aeolesthes sarta* attack in two major cities of Pakistan based on survey respondents' opinion.

| Variables | Coefficients | *t*-Stat |
|---|---|---|
| **Independent variables** | | |
| Gender [a] | 0.09 *** | 3.54 |
| Age [b] | 0.02 | 0.97 |
| Designation [c] | −0.27 *** | −11.11 |
| Experience [d] | 0.20 *** | 7.11 |
| Location [e] | −0.11 *** | −4.21 |
| Forest Value [f] | 0.54 *** | 19.45 |
| Knowledge about pest [g] | 0.08 ** | 2.62 |
| Personal satisfaction towards control program [h] | −0.56 *** | −19.15 |
| Federal/Local govt. control fund program [i] | −0.09 *** | −3.47 |
| Stand suitable for attack [j] | 0.23 *** | 7.29 |
| Type of Tree species mainly attack [k] | 0.46 ** | 2.03 |
| The average number of trees attacked/damaged [l] | 0.54 *** | 20.23 |

**Table 5.** *Cont.*

| Variables | Coefficients | *t*-Stat |
|---|---|---|
| **Regression model** | | |
| N | 399 | |
| *F*-statistic | 130.89 *** | |
| Adjusted $R^2$ | 0.66 | |

***, ** Coefficient significance estimates at the $p < 0.001$, and $p < 0.01$ level. [a] Dummy variable: 1 = male, and 0 = female. [b] Dummy variable: 1 = age is >45 y, and 0 = age is <45 y. [c] Dummy variable: 1 = government forest department worker, and 0 = others. [d] Dummy variable: 1 = >15 y, and 0 = <15 y. [e] Dummy variable: 1 = Peshawar, and 0 = Quetta. [f] Variable generated by summing up all benefits and uses of forests. [g] Dummy variable: 1 = yes to the statement "Do you know that SLB causes tree internal damage and often kills trees?", and 0 = no. [h] Dummy variable: 1 = yes, and 0 = no. [i] Dummy variable: 1 = funding amount for SLB control program was >100,000$, and 0 = <100,000$. [j] Dummy variable: 1 = pure stand, and 0 = mixed or both. [k] Dummy variable: 1 = *Populus* spp., and 0 = other species. [l] Dummy variable: 1 = >150 trees/ha, and 0 = <150 trees/ha.

### 3.5. Priorities over Measures for Pest Control and Forest-Type Protection

When asked to evaluate various control options used for SLB outbreaks, respondents from both locations preferred a biological control agent (*Beauveria bassiana*) (Figure 5a). This was followed by biological insecticide, synthetic chemical control, and silvicultural treatments for SLB for QU respondents. For PE respondents, biological control agents were preferred, followed by biological insecticide, silvicultural treatment, and synthetic chemical control for SLB. PE residents had a statistically higher prioritization percentage of silvicultural treatments and a lower percentage of synthetic chemical use than QU residents.

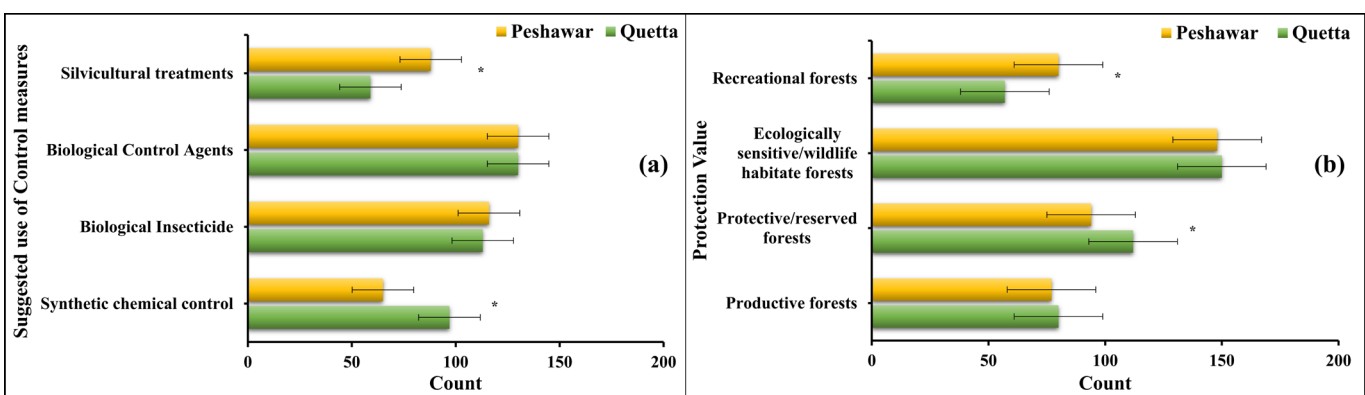

**Figure 5.** Respondent's preference concerning (**a**) control options used for SLB, (**b**) and forest type that should be protected for SLB outbreak in Quetta and Peshawar. Bars represent frequencies ± SE. * Indicating that frequencies were statistically different between cities at $p < 0.05$ (chi-square test).

Respondents from both cities (QU = 38%, PE = 37%) stated that ecologically sensitive areas and wildlife habitats should be prioritized for SLB outbreaks (Figure 5b). Protective/reserved forests followed these areas as productive forests and recreation forest areas for SLB outbreaks in both cities. However, a significantly higher percentage of respondents from QU (28%) compared to PE (24%) stated that protective/reserved forests should be protected as a second priority. In QU, productive forest areas were placed above recreational forests, whereas PE recreation areas prioritized more than productive forests. Similarly, there were statistical variations in the rankings of cities for forest-type protection. Specifically, QU respondents ranked much higher in protecting protective/reserved forests against SLB outbreaks than PE respondents. In addition, PE respondents ranked much higher than QU respondents in preserving recreational forests against SLB outbreaks.

### 3.6. Respondent's Attitude on Pest Control, Payer Options, and the Scope of the Control Program

According to a substantial percentage of respondents from QU (95%) and PE (92%), the regional or federal government should support development and research to enhance

biological control techniques in order to avert forest losses related to pests (Table 6). There was also strong support (QU = 90% − PE = 89%) regarding controlling future SLB outbreaks. When it came to preferences for funding for research and development, there were considerable differences between cities but not for support for potential SLB control. Of those respondents favoring control of future SLB outbreaks, 46% of respondents from QU and 44% of respondents from PE thought that the provincial/federal government should pay control costs on forests and woodlands, whereas 42% of respondents from QU and 43% of respondents from PE thought that both provisional/federal government and land owners combinedly should pay control costs on government or private woodland forests (Table 6). However, statistically, more QU respondents (14% vs. 10%) stated that land owners and private organizations should pay the control costs on public and privately owned woodland forests. For the next SLB outbreak, respondents from both cities who supported outbreak control predominantly favored protecting more forested areas (above 58%) (Table 6). QU respondents who supported pest control suggested statistically greater forest area protection from SLB outbreaks than PE respondents (61% vs. 58%).

**Table 6.** Survey respondents' satisfaction and attitude to pest control, pest control costs payer, and protection scope for future SLB outbreaks.

| Attitudes | Quetta | | Peshawar | |
|---|---|---|---|---|
| | N | % [a] | N | % [a] |
| **Attitude towards pest control:** | | | | |
| Support for financing for pest control research and development | 359 | 95.42 * | 348 | 91.50 * |
| Support for controlling future *SLB* outbreaks | 357 | 90.31 | 344 | 89.67 |
| **Who should pay the cost of the SLB control program on forest land and privately owned wood and farmlands? [b]** | | | | |
| The provisional/Federal Government should pay | 324 | 46.18 * | 309 | 44.03 * |
| Landowners + private organizations should pay | 324 | 14.17 * | 309 | 10.54 * |
| Both (50/50) should pay | 324 | 41.79 * | 309 | 43.27 * |
| Percentage of forest land that should be protected from *SLB* outbreak. [b,c] | 309 | 61.25 (0.92) ‡ | 301 | 58.05 (1.08) ‡ |

* Demonstrated that distributions between cities are statistically different at $p < 0.05$ (Chi-square test). ‡ Demonstrated that distribution means between cities are statistically different at $p < 0.05$ (Mann-Whitney-Wilcoxon U *t*-test). [a] Percentage of respondents chose "yes" when asked a question. [b] Statistics derived from respondents who favored preventing future SLB outbreaks. [c] Mean % ± (standard error).

Regression analysis of features persuading public preference over areas that should be protected from future SLB outbreaks in their cities is conferred in Table 7. Overall, the coefficients of the regression model were statistically significant. All the variables positively influenced public preference for protected areas except age and location. Those who were males, those who had 15 years of experience and more, those who had rated the forest value high, those who had pest knowledge, those who supported future SLB control programs, those who preferred the use of biological control agents for controlling SLB outbreak, those who preferred protection of ecological sensitive/wildlife forests, those who stated the government should pay the cost, and those who stated forest survey should be conducted once in a year preferred significantly more area to be protected. However, those below 45 years of age and those who were QU residents preferred significant areas to be protected. The F-statistic value in regression (F = 418.70, $p < 0.001$) demonstrated that the null hypothesis of no association between the dependent and independent variables could be reliably ignored. In general, the variables featured in the regressions outlined 86% of the variability concerning preferred area protection.

**Table 7.** Regression results of factors influencing people's attitude towards future outbreaks of *Aeolesthes sarta* in two major cities of Pakistan based on survey respondents' opinions.

| Variables | Coefficients | *t*-Stat |
|---|---|---|
| **Independent variables** | | |
| Gender [a] | 0.07 *** | 4.89 |
| Age [b] | −0.02 * | −2.97 |
| Designation [c] | 0.24 *** | 12.01 |
| Experience [d] | 0.16 *** | 9.33 |
| Location [e] | −0.12 *** | −7.58 |
| Forest Value [f] | 0.81 *** | 51.12 |
| Knowledge about pest [g] | 0.05 * | 2.41 |
| Support for future SLB control program [h] | 0.68 *** | 32.26 |
| Preferred control methods [i] | 0.11 *** | 6.92 |
| Preferred forest type to be protected [j] | 0.52 *** | 25.27 |
| Who should pay the control program cost [k] | 0.09 *** | 5.55 |
| How often should field surveys be conducted [l] | 0.39 *** | 16.99 |
| **Regression model** | | |
| N | 399 | |
| *F*-statistic | 418.70 *** | |
| Adjusted $R^2$ | 0.86 | |

***, * Coefficient significance estimates at the $p < 0.001$, and $p < 0.05$ levels. [a] Dummy variable: 1 = male, and 0 = female. [b] Dummy variable: 1 = age is >45 y, and 0 = age is <45 y. [c] Dummy variable: 1 = government forest department worker, and 0 = others. [d] Dummy variable: 1 = >15 y, and 0 = <15 y. [e] Dummy variable: 1 = Peshawar, and 0 = Quetta. [f] Variable created by summing all forest uses and benefits. [g] Dummy variable: 1 = yes to the statement "Do you know that *SLB* causes tree internal damage and often kills trees?", and 0 = no. [h] Dummy variable: 1 = yes, and 0 = no. [i] Dummy variable: 1 = biological control agent, and 0 = others (synthetic chemical control, biological insecticide, silvicultural treatments). [j] Dummy variable: 1 = ecological sensitive/wildlife forests, and 0 = others (productive forests, protective/reserved forests, recreational forests). [k] Dummy variable: 1 = government, and 0 = others (land owners/private organizations). [l] Dummy variable: 1 = once in a year, and 0 = others (once in two years, once in three years, once in five years).

## 4. Discussion

The literature review revealed that no studies on the social aspects of natural disturbance in natural or artificial forests had been conducted in Pakistani territory, and no studies on public perceptions of pest infestation and its management in forests have been found. This study bridges that gap by investigating local citizens' perspectives of SLB damage cost and management in Quetta (QU) and Peshawar (PE) in northwestern Pakistan.

There were significant differences in public opinion regarding the cost of pest damage in QU and PE. Variations occurred in the following order: (i) kind of stand that sustained the most damage; (ii) type of tree species that have been attacked the most; (iii) average number of trees/ha damaged annually; (iv) government funding amount for SLB control program; and (v) approximate damage cost due to SLB outbreak. In particular, opinions regarding the average number of trees damaged annually by SLB differed significantly across cities. Although the *Populus* species most commonly attacked by SLB were scored high in both cities, QU respondents valued *Populus* species much more than PE. Furthermore, PE residents evaluated pure stands as more susceptible to SLB attack than mixed/both stands, whereas QU respondents scored these in reverse order. QU and PE are two of the main cities of Pakistan that have suffered from intense outbreaks of SLB [15,24–27], so people may be more familiar with the pest, its nature of attack/damage, and its host preferences. However, QU's damage frequency was higher than PE's [3].

Generally, respondents' satisfaction rate towards the government funding and SLB control program did influence the respondents' opinion about SLB damage cost, especially QU respondents who were least satisfied with the government programs to control SLB. Consistent with our hypotheses, greater knowledge of the SLB, type of stand damaged, type of tree species damaged, average number of trees damaged, gender, and more experience were associated with stating significantly higher damage costs due to SLB attack. While the

cost of damage was significantly different between both cities, the infect value was greater for QU versus PE. The lesser the personal satisfaction towards government pest control programs, the higher the damage cost [38]. Residents with more experience and those who worked for the government forest department stated low damage costs. Contrary to our hypotheses, age did not influence respondents' opinions about damage cost. QU respondents stated the highest mean value as a pest damage cost than PE; these findings are supported by the previously reported outbreaks and their greater intensities in QU than PE [3,25,26].

In QU and PE, there were a number of significant and distinct variations in public perspectives concerning insect outbreaks and control options. The following attitudes varied: (i) preferences for control options; (ii) priorities for protecting particular forest types; (iii) who should foot the bill for control on both privately and government-owned forestland; (iv) frequency of forest surveys; and (v) preferred percentage of forest area to be defended from future SLB outbreaks. The preference for silvicultural treatments and synthetic chemical control for SLB differed significantly between cities. Despite being evaluated the lowest in both cities, QU respondents preferred synthetic chemical control over PE. Furthermore, PE residents valued silvicultural treatments higher than QU. These findings contrast with [28].

QU is the capital city of Pakistan's Baluchistan province. It is known as the "Fruit Garden of Pakistan", with the majority of the area in farmland [29], so the utilization of synthetic chemical pesticides may be more widely known and accepted by the general public; (2) in QU, the use of synthetic chemicals for SLB control was very frequent [24], so people may be more adaptive to the use of synthetic chemical control. There was no literature source found that highlighted the encountering of public opposition regarding tree protection measures against the pest in both cities; however, we strongly recommend that there is a need to implement comprehensive educational campaigns tailored to the QU community, emphasizing the risks associated with synthetic pesticides and highlighting the benefits of biological alternatives. Incorporating interactive workshops informational sessions and distributing educational materials can enhance awareness and promote the adoption of sustainable pest management practices [39,40]. Some studies reported that in developing nations, overuse and misuse of pesticides stem from limited education on alternatives, scant hazard awareness, market demands for flawless crops, and farmer aversion to risks [41–43]. Education correlates with improved pesticide handling and access to safety information [44], whereas less-educated farmers face barriers in acquiring and adhering to safety guidelines [45].

Respondents from QU regarded protecting forest land as a higher priority than PE respondents in protecting recreation forests from an outbreak of SLB. It could be because many forest areas in QU come under the category of historical forest and are protected by the government [46], and people are very much aware of their value. Forests' ecological, recreational, and production values significantly shape corresponding attitudes towards forest management. Ecological values drive an environmental perspective, while recreational values prioritize human-centered management. Production values dictate an economically focused approach to forest management, reflecting distinct priorities and orientations within the broader framework of sustainable forest stewardship [47]. However, in PE, no historical forests are present that need to be protected, while some natural and planted forests are valued as reserved forests guarded by the local forest department. To bolster awareness of the value of forests among PE individuals, we advocate for immersive educational programs, community engagement initiatives, and forest conservation campaigns. Incorporating experiential learning activities, such as guided forest walks and interactive workshops, can cultivate a deeper appreciation for forests and their ecological significance.

Another variation in attitudes among respondents concerned who should pay for pest control costs. Significantly more QU than PE respondents supported the idea that the province government pay for pest control on public and privately owned woodlands.

Furthermore, significantly more QU respondents than PE respondents preferred that landowners/private organizations pay for the pest control program. This attitude could be because QU has more privately owned farmlands than PE [29].

Finally, in this investigation, several criteria that strongly influenced the preferred proportion of land that should be protected from future outbreaks were in line with our hypothesis; for example, we efficiently hypothesized that those who were males, had higher levels of education, had more experience, supported future SLB control programs, preferred biological control, preferred ecologically sensitive forest protection, supported government payment for control programs, and preferred yearly-based survey favored a more significant percentage of forest area protected from SLB outbreaks. According to our hypotheses, all variables significantly influence respondents' preference to protect a greater percentage of the area from SLB outbreaks. Furthermore, age was negatively correlated with the preferred percentage of area to be protected from future SLB outbreaks. This finding is in line with the findings of McFarlane et al. [38]; they found that senior citizens were less inclined to support efforts to control mountain pine beetle infestations in two Western Canadian National Parks. Those with SLB knowledge, as expected, preferred more areas to be protected from future SLB outbreaks. This finding contrasts with Chang et al. [28], who said that pest knowledge significantly influenced public preference for pest control extent. We underscore the imperative of promoting enhanced educational training among females to heighten awareness about forest protection and recommended control measures. Tailored programs should focus on empowering women with knowledge of sustainable practices. Simultaneously, addressing the behavior of older individuals in pest control programs involves targeted awareness campaigns, emphasizing the benefits of eco-friendly methods. Integrating their traditional wisdom with contemporary approaches fosters a collaborative and effective pest management strategy, aligning with cultural practices and environmental conservation goals.

Finally, contrary to our hypothesis, the city of residence impacted preferences over the percentage of protected area. Residents from QU preferred a higher percentage of areas protected from future SLB outbreaks than PE. There may be some overlap between this and the previously mentioned reasons that explain why cities differ in their control preferences. This finding is in line with Chang et al. [28], who found that the location of the residents significantly influenced the extent of the pest control.

The question of whether information about insect infestation attributes influences people to support pest control is debatable. Molnar et al. [48] and McFarlane and Witson [49] concluded that simply presenting data on forest pest infestations does not always result in support for pest control. Moreover, MacDonald et al. [50] stated that the respondents' unique encounters with pest-related problems had no appreciable impact on their attitudes about pest control, coming to the conclusion that "people regard all insect pests identically—a bug is a bug is a bug". However, our data show that most people's familiarity with SLB alters their preference for the scope of control.

Compared to less knowledgeable people, those more aware of the pest stated 12% higher damage costs and favored protecting 16% more forest area from SLB outbreaks. Chang et al. [28] claimed that individuals who were more knowledgeable about pests favored 15% and 8% of forest land protected from *Choristoneura fumiferana* and *Malacosoma disstria* outbreaks, respectively. The fact that those who know about pest attributes want a greater proportion of forest protected from pests that do cause greater harm (i.e., SLB) gives plausibility to the theory that information might impact choice of control extent. These results support MacDonald et al.'s [51] claim that "if resource executives want the public to evaluate pest control alternatives more precisely, public relations efforts should likely emphasize the ecological and economic contrasts of insect". As a result, by making such evidence available to the public, policymakers and forest managers will be more qualified to develop publicly permissible pest control strategies.

Most of the study findings are supported by other research literature. Notably, we demonstrated that the majority of respondents in both cities (1) anticipated high pest

damage costs; (2) supported biological control of future SLB outbreaks so that pest damage costs stated by respondents in both cities could be minimized; (3) considered that to reduce pest-related forest losses, the federal and provincial governments should support research and development to advance biological pest control strategies; and (4) placed a higher priority on protecting ecologically vulnerable areas and wildlife habitat. These findings are congruent with those of Wagner et al. [52] and McFarlane et al. [38], who stated that the public in several regions of North America was far more inclined to support biological control than chemical control. According to McFarlane et al. [38], the general public in two Western Canadian inhabitants close to National Parks perceived pest outbreaks as detrimental to the ecology and believed pest control techniques would help the areas' ecological stability.

The results demonstrating that QU residents were statistically more aware of prior SLB outbreaks than PE residents coincide with the findings of MacDonald et al. [51], Flint [53], and Chang et al. [28], who stated that the general public tends to be more familiar and well conversant with forest pests that affect neighborhoods somewhat severely. The QU and PE public have suffered notable SLB outbreaks [3]. Unsurprisingly, QU residents were better informed about SLB than PE residents.

Living in QU, for example, is a factor connected with social dependency on natural resources, and these factors appear positively related to the recommended percentage protection area from the SLB outbreak. This supports the research findings of Flint [53] and McFarlane and Witson [49] that residents scrutinized forest pest influences distinctly based on their contact with the pest as well as their possible social, economic, and biophysical effects. These results reinforce Brunson and Shindler's [54] contention that using generalized natural disturbance strategies or information approaches is ineffective. Therefore, for future sustainable pest control plans, it is essential to comprehend variations in location-specific environmental and social elements and personal attitudes.

## 5. Conclusions

This study has significantly highlighted the people's perspective on pest damage cost due to the SLB outbreak and their attitude toward future control programs for controlling SLB outbreaks in two cities in Pakistan. We found that the rate of damaged trees was high in QU compared to PE, and because of that, pest damage cost was also high in QU compared to PE. Multivariant model results indicated that the respondent from QU indicated a high damage cost compared to PE. Additionally, we found that people preferred biological control methods, ecologically sensitive areas to be protected, the government to pay the cost for future control and funding programs, and forest surveys to be conducted once a year. Multivariant model results indicated that the respondents from QU preferred more percentage of the area to be protected from future SLB outbreaks than PE. Incorporating public preferences reported in this study into pest control decision-making processes might represent one of the best approaches to increasing public participation in natural resource management. We reckon that doing so would considerably increase policy efficacy by reducing possible disputes between forest managers and general population members.

**Author Contributions:** Conceptualization, U.H. and A.A.; methodology, U.H.; software, U.H. and A.A.; validation, U.H. and A.A.; formal analysis, U.H.; investigation, U.H.; resources, U.H.; data curation, U.H. and A.A.; writing—original draft preparation, U.H.; writing—review and editing, U.H. and A.A; visualization, U.H.; supervision, J.S.; project administration, J.S.; funding acquisition, J.S. All authors have read and agreed to the published version of the manuscript.

**Funding:** This study was supported by the Forestry Science and Technology Innovation Special of Jiangxi Forestry Department (201912) and the National Natural Science Foundation of China (32171794).

**Data Availability Statement:** Data will be available upon request.

**Acknowledgments:** The authors thank Abubakar Sadique Ibrahim for his assistance. Also, thanks to Quetta and Peshawar local governments for helping to conduct this research.

**Conflicts of Interest:** The authors declare no conflicts of interest.

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
