# Peer review of "Public Attitudes towards Forest Pest Damage Cost and Future Control Extent: A Case Study from Two Cities of Pakistan"

_forests, doi:10.3390/f15030544_

Round 1
Reviewer 1 Report
Comments and Suggestions for Authors
The paper titled "Public attitude towards forest pest damage cost and future control extent; a case study from two cities of Pakistan" presents a very interesting insight into the perceptions of two localities in Pakistan regarding damage caused by a coleopteran pest. Overall, it is a well-written piece in English and also features a very compelling and powerful statistical analysis for examining survey data. The use of dummy variables as well as the utilization of chi-square homogenization tests seems appropriate. The results are clear and consistent with the discussion.
Some general considerations regarding English: perhaps there is the use of overly formal words or terms like "annum." There's no problem with using "year," although if it's the authors' style, it's not an issue.
Additionally, here are some considerations regarding the manuscript: 1) In the abstract, in the third line, it mentions the tree species affected by the coleopteran. However, when these are named, only the genera are mentioned. In short, I believe they should be written as "sp" or alternatively, specify which species, or simply define that they correspond to genera and not species.
2) In line 125, the authors use "(sq mi)" as a unit, which is fine, but they have already mentioned the equivalent in km2. It's unnecessary to include all units; they should stick to just km2. This should be standardized throughout the text. Similarly, in line 126, they mention 3470m and then in parentheses "m.s.l." Just retain "m.s.l."
3) Figures 1 and 2 should be added with improved resolution, especially regarding the text.
4)In lines 180 and 181, the same situation occurs as in the abstract regarding the fact that tree species are not named but rather their genera.
Author Response
Thank you for bringing attention to our manuscript revisions. We have meticulously incorporated feedback from you, ensuring comprehensive improvements throughout the manuscript. Please refer to the details outlined in the main text file. We are confident that this revised version adequately addresses all queries and enhances the quality of our work.
Some general considerations regarding English: perhaps there is the use of overly formal words or terms like "annum." There's no problem with using "year," although if it's the authors' style, it's not an issue.
Answer: Thank you for clarifying. We will maintain the current format as per your preference.
Additionally, here are some considerations regarding the manuscript:
Question: In the abstract, in the third line, it mentions the tree species affected by the coleopteran. However, when these are named, only the genera are mentioned. In short, I believe they should be written as "sp" or alternatively, specify which species, or simply define that they correspond to genera and not species.
Answer: Thank you for highlighting. Comment has been addressed as suggested.
Question: In line 125, the authors use "(sq mi)" as a unit, which is fine, but they have already mentioned the equivalent in km2. It's unnecessary to include all units; they should stick to just km2. This should be standardized throughout the text. Similarly, in line 126, they mention 3470m and then in parentheses "m.s.l." Just retain "m.s.l."
Answer: Thank you for highlighting. Comment has been addressed as suggested please see the details in revised MS.
Question: Figures 1 and 2 should be added with improved resolution, especially regarding the text.
Answer: Thank you for highlighting. We will provide the high-resolution images during resubmission of revised version.
Question: In lines 180 and 181, the same situation occurs as in the abstract regarding the fact that tree species are not named but rather their genera.
Answer: Thank you for highlighting. Comment has been addressed as suggested.
Reviewer 2 Report
Comments and Suggestions for Authors
In this research, the authors carry out a study through surveys to know the perception of the population of two cities in Pakistan regarding the damage caused by Aeolesthes Sarta, the cost of this damage, the level of satisfaction with the government's measures to mitigate the damage, the control strategies that should be employed and the magnitude of the areas that should be protected in future outbreaks. This work represents an interesting contribution by focusing on the perception that citizens have about a pest problem, its importance, management, and knowledge about it. As the authors refer, it is also a reference for designing policies with a participatory approach focused on communities.
On the other hand, there are several aspects of the work that should be reviewed and improved for publication:
The authors must reconsider whether the t-test is the correct choice in some of the discrete scale variable analyses.
Also, there are aspects of the discussion that can be expanded.
Specific comments are listed below:
Line 17: "Trees (181 + 1.20 trees/ha/annum) were damaged" Did you mean 181±1.20?
Line 19 "QU 19 (480840.80 + 4716.94$/annum) compared to PE" did you mean 480840±4716.94?
Line 38 "families (Rossa and Goczal, 2021)[2]": Authors and years should not be entered. Only the reference number. This must be corrected throughout the text.
Line 41: "such as Populus species, Juglans species, Acer species, Salix species, Malus species, Platanus species, and Ulmus species" change by "belonging mainly to the genus Populus, Juglans, Acer..."
Line 66 and 134: ">1800m m.s.l" change by ">1800 m.a.s.l.
Line 129: " 32987ha (329.87 km2)" "32987 ha" If the area is expressed in ha, it is not necessary to express it again in km2
Lines 133 and 137: "of 1,257 km2 (485 sq mi)" Unify units: km2, Ha, square miles? and use only one.
Line 135: "receives pre-135 cipitation in the early and late months of the year" July and August are not late months of the year.
Line 239: "Independent sample t-tests were used for investigating variations in city means.": Because these are discrete scores on a scale, it cannot be assumed that the data have a normal distribution, as is appropriate for a Student's t-test. In this case, it would be more appropriate to use the Mann-Whitney test.
Figure 3 legend: Standard errors were 0.03 for all variables?? Because these are discrete scores on a scale, it cannot be assumed that the data have a normal distribution, as is appropriate for a Student's t-test. In this case, it would be more appropriate to use the Mann-Whitney test.
Line 337: "responses from respondents from both cities were nearly identical" change by "responses from respondents relative to the origin of the pest from both cities were nearly identical"
Lines 348, 356, 359, 360, and Figure 4: List the different graphs included in Figure: 4a, 4b, 4c, and 4d, and cite in the text the graph that is specifically being interpreted. Example (Figure 4a).
Lines 351, 352, and 354: Italics for scientific names
Table 4: "What is the average funding amount for pest control in your city?" and "What is the approximate forest damage cost?": Because these are discrete scores on a scale, it cannot be assumed that the data have a normal distribution, as is appropriate for a Student's t-test. In this case, it would be more appropriate to use the Mann-Whitney test.
Lines 416, 423, and figure 5: List the different graphs included in figure: 5a,5b, and cite in the text the specifically interpreted graph. Example (Figure 5a)
Line 418: "For PE respondents, biological control agents were used by biological insecticide" change to "For PE respondents, biological control agents were preferred, followed by biological insecticide"
Line 420: "PE residents had a statistically higher percentage of silvicultural treatments" change by "PE residents had a statistically higher prioritization percentage of silvicultural treatments"
Table 6: "Percentage of forest land that should be protected from SLB outbreak" Because these are discrete scores on a scale, it cannot be assumed that the data have a normal distribution, as is appropriate for a Student's t-test. In this case, it would be more appropriate to use the Mann-Whitney test.
Table 7. "Regression results of factors influencing people's attitude towards future outbreaks of Aeolesthes sarta in two major cities of Pakistan based on survey respondent´s opinions. " How was this independent variable obtained? Is it a weighting of the proportions presented in the first two questions of Table 6 and the last one? Consider only the percentage of forest land that should be protected?
Line 538....farmland (PBS, 2017)[29], so the utilization of synthetic chemical pesticides may be more widely known and accepted by the general public": What do the authors propose to ensure that QU people have greater clarity about the risks and disadvantages of synthetic pesticides, and recognize the value of biological strategies?
Line 546: "However, in PE, no such historical forests are present that need to be protected, while some natural and planted forests are valued as reserved forests guarded by the local forest department": If PE people are not as aware of the value of forests, what do the authors propose to increase this awareness?
Line 557: "We efficiently hypothesized that those who were males had higher levels of education, had more experience, supported future SLB control programs, preferred biological control, preferred ecologically sensitive forest protection, supported government payment for control programs, and preferred yearly-based survey favored a more significant percentage of forest area protected for SLB outbreaks": The authors must recognize in the text that this raises the need to carry out activities aimed at promoting a higher level of educational training in the female population and thus achieve greater awareness about the need to protect forests, the control measures that are recommended, etc
563: "Furthermore, age was negatively correlated with the preferred percentage of area to be protected from future SLB outbreaks.": Could it then be concluded that senior citizens are less aware of the importance of these protective measures? What actions can be taken to raise awareness among this group about the importance of forests and their protection?
Additional editing changes can be found in the attached file.

Comments on the Quality of English LanguageMinor English editing changes required
Author Response
Thank you for bringing attention to our manuscript revisions. We have meticulously incorporated feedback from you, ensuring comprehensive improvements throughout the manuscript. Please refer to the details outlined in the main text file. We are confident that this revised version adequately addresses all queries and enhances the quality of our work.
Question: The authors must reconsider whether the t-test is the correct choice in some of the discrete scale variable analyses.
Answer: Thank you for addressing this matter. Following your suggestion, we conducted discrete scale variable analyses using the Mann-Whitney-Wilcoxon U t-test. Accordingly, as per your recommendation, we have replaced the independent sample Student’s t-test with Mann-Whitney-Wilcoxon U t-test.
Question: Also, there are aspects of the discussion that can be expanded.
Answer: Thank you for bringing attention to our manuscript revisions. We have meticulously incorporated feedback from you, ensuring comprehensive improvements in the discussion section. Please refer to the details outlined in the discussion section in main text file. We are confident that this revised version adequately addresses all queries and enhances the quality of our work.
Specific comments are listed below:
Question: Line 17: "Trees (181 + 1.20 trees/ha/annum) were damaged" Did you mean 181±1.20?
Answer: Thank you for highlighting. Comment has been addressed as suggested.
Question: Line 19 "QU 19 (480840.80 + 4716.94$/annum) compared to PE" did you mean 480840±4716.94?
Answer: Thank you for highlighting. Comment has been addressed as suggested.
Question: Line 38 "families (Rossa and Goczal, 2021)[2]": Authors and years should not be entered. Only the reference number. This must be corrected throughout the text.
Answer: Thank you for highlighting. Comment has been addressed as suggested throughout the MS.
Question: Line 41: "such as Populus species, Juglans species, Acer species, Salix species, Malus species, Platanus species, and Ulmus species" change by "belonging mainly to the genus Populus, Juglans, Acer..."
Answer: Thank you for highlighting. Comment has been addressed as suggested.
Question: Line 66 and 134: ">1800m m.s.l" change by ">1800 m.a.s.l.
Answer: Thank you for highlighting. Comment has been addressed as suggested.
Question: Line 129: " 32987ha (329.87 km2)" "32987 ha" If the area is expressed in ha, it is not necessary to express it again in km2
Answer: Thank you for highlighting. Comment has been addressed as suggested.
Question: Lines 133 and 137: "of 1,257 km2 (485 sq mi)" Unify units: km2, Ha, square miles? and use only one.
Answer: Thank you for highlighting. Comment has been addressed as suggested.
Question: Line 135: "receives pre-135 cipitation in the early and late months of the year" July and August are not late months of the year.
Answer: Thank you for highlighting. Comment has been addressed as suggested.
Question: Line 239: "Independent sample t-tests were used for investigating variations in city means.": Because these are discrete scores on a scale, it cannot be assumed that the data have a normal distribution, as is appropriate for a Student's t-test. In this case, it would be more appropriate to use the Mann-Whitney test.
Answer: Thank you for addressing this matter. Following your suggestion, we conducted discrete scale variable analyses using the Mann-Whitney-Wilcoxon U t-test. Accordingly, as per your recommendation, we have replaced the independent sample Student’s t-test with Mann-Whitney-Wilcoxon U t-test.
Question: Figure 3 legend: Standard errors were 0.03 for all variables?? Because these are discrete scores on a scale, it cannot be assumed that the data have a normal distribution, as is appropriate for a Student's t-test. In this case, it would be more appropriate to use the Mann-Whitney test.
Answer: Thank you for addressing this matter. Following your suggestion, we conducted discrete scale variable analyses using the Mann-Whitney-Wilcoxon U t-test. Accordingly, as per your recommendation, we have replaced the independent sample Student’s t-test with Mann-Whitney-Wilcoxon U t-test.
Question: Line 337: "responses from respondents from both cities were nearly identical" change by "responses from respondents relative to the origin of the pest from both cities were nearly identical"
Answer: Thank you for highlighting. Comment has been addressed as suggested.
Question: Lines 348, 356, 359, 360, and Figure 4: List the different graphs included in Figure: 4a, 4b, 4c, and 4d, and cite in the text the graph that is specifically being interpreted. Example (Figure 4a).
Answer: Thank you for highlighting. Comment has been addressed as suggested.
Question: Lines 351, 352, and 354: Italics for scientific names
Answer: Thank you for highlighting. Comment has been addressed as suggested.
Question: Table 4: "What is the average funding amount for pest control in your city?" and "What is the approximate forest damage cost?": Because these are discrete scores on a scale, it cannot be assumed that the data have a normal distribution, as is appropriate for a Student's t-test. In this case, it would be more appropriate to use the Mann-Whitney test.
Answer: Thank you for addressing this matter. Following your suggestion, we conducted discrete scale variable analyses using the Mann-Whitney-Wilcoxon U t-test. Accordingly, as per your recommendation, we have replaced the independent sample Student’s t-test with Mann-Whitney-Wilcoxon U t-test.
Question: Lines 416, 423, and figure 5: List the different graphs included in figure: 5a,5b, and cite in the text the specifically interpreted graph. Example (Figure 5a)
Answer: Thank you for highlighting. Comment has been addressed as suggested.
Question: Line 418: "For PE respondents, biological control agents were used by biological insecticide" change to "For PE respondents, biological control agents were preferred, followed by biological insecticide"
Answer: Thank you for highlighting. Comment has been addressed as suggested.
Question: Line 420: "PE residents had a statistically higher percentage of silvicultural treatments" change by "PE residents had a statistically higher prioritization percentage of silvicultural treatments"
Answer: Thank you for highlighting. Comment has been addressed as suggested.
Question: Table 6: "Percentage of forest land that should be protected from SLB outbreak" Because these are discrete scores on a scale, it cannot be assumed that the data have a normal distribution, as is appropriate for a Student's t-test. In this case, it would be more appropriate to use the Mann-Whitney test.
Answer: Thank you for addressing this matter. Following your suggestion, we conducted discrete scale variable analyses using the Mann-Whitney-Wilcoxon U t-test. Accordingly, as per your recommendation, we have replaced the independent sample Student’s t-test with Mann-Whitney-Wilcoxon U t-test.
Question: Table 7. "Regression results of factors influencing people's attitude towards future outbreaks of Aeolesthes sarta in two major cities of Pakistan based on survey respondent´s opinions. " How was this independent variable obtained? Is it a weighting of the proportions presented in the first two questions of Table 6 and the last one? Consider only the percentage of forest land that should be protected?
Answer: This variable was obtained by considering only the percentage of forest land that should be protected. For this, we used the following question because our focus was to highlight the area that should be covered in future outbreaks.
Question: Line 538....farmland (PBS, 2017)[29], so the utilization of synthetic chemical pesticides may be more widely known and accepted by the general public": What do the authors propose to ensure that QU people have greater clarity about the risks and disadvantages of synthetic pesticides, and recognize the value of biological strategies?
Answer: Thank you for notifying us. The comment has been duly addressed as per your suggestion. We have included a brief passage in the discussion section to elaborate on this matter. For further details, please refer to the discussion section of the document.
Question: Line 546: "However, in PE, no such historical forests are present that need to be protected, while some natural and planted forests are valued as reserved forests guarded by the local forest department": If PE people are not as aware of the value of forests, what do the authors propose to increase this awareness?
Answer: Thank you for notifying us. The comment has been duly addressed as per your suggestion. We have included a brief passage in the discussion section to elaborate on this matter. For further details, please refer to the discussion section of the document.
Question: Line 557: "We efficiently hypothesized that those who were males had higher levels of education, had more experience, supported future SLB control programs, preferred biological control, preferred ecologically sensitive forest protection, supported government payment for control programs, and preferred yearly-based survey favored a more significant percentage of forest area protected for SLB outbreaks": The authors must recognize in the text that this raises the need to carry out activities aimed at promoting a higher level of educational training in the female population and thus achieve greater awareness about the need to protect forests, the control measures that are recommended, etc
Answer: Thank you for notifying us. The comment has been duly addressed as per your suggestion. We have included a brief passage in the discussion section to elaborate on this matter. For further details, please refer to the discussion section of the document.
Question: 563: "Furthermore, age was negatively correlated with the preferred percentage of area to be protected from future SLB outbreaks.": Could it then be concluded that senior citizens are less aware of the importance of these protective measures? What actions can be taken to raise awareness among this group about the importance of forests and their protection?
Answer: Thank you for notifying us. The comment has been duly addressed as per your suggestion. We have included a brief passage in the discussion section to elaborate on this matter. For further details, please refer to the discussion section of the document.
Reviewer 3 Report
Comments and Suggestions for Authors
This manuscript is well written; however, some sentences are very long. It is hard to understand for the reader.
Abstract: Pls – rewrite in some sentences (see the attached file)
Introduction: minor comments (see the attached file)
Results and discussion: minor comments

Author Response
Thank you for bringing attention to our manuscript revisions. We have meticulously incorporated your feedback, ensuring comprehensive improvements throughout the manuscript. Please refer to the details outlined in the main text file. We are confident that this revised version adequately addresses all queries and enhances the quality of our work.
We appreciate your suggestion for the study area (Materials and Methods section), but several studies presented the area's geographical description in the materials and methods sections, i.e., Chang et al., 2009.
Besides that, we have included a separate zip file of figures so that high-resolution images can be published in the final online MS.
Reviewer 4 Report
Comments and Suggestions for Authors
The manuscript, entitled 'Public Attitude Towards Forest Pest Damage Cost and Future Control Extent; A Case Study from Two Cities of Pakistan', raises an interesting issue regarding the viewpoints of different sections of society on the mass occurrence of the pest Aeolesthes sarta (Coleoptera: Cerambycidae) in their area, tree health and strategies to protect trees from various similar pests. The paper is written in an understandable language. The presentation of the results is clear. Minor editorial errors (e.g. verse 119 - two steps at the end of a sentence) need to be corrected. Although the manuscript seems interesting and valuable, several questions arise that should be addressed in more detail to justify the purpose of the survey. These are:
1) To what extent does Aeolesthes sarta threaten stands today in the study area? The publications presented on this topic mainly refer to observations made in the 20th century (Kulinich, 1965, Krivosheina, 1984, Gaffar and Bhat, 1991, Orlinski et al. 1991) and early 21st century (Farashiani et al. 2001, EPPO, 2005).
2) Were and what methods of protecting trees from Sart longhorned beetle A. sarta carried out in the study area?
3) Have the green space management services and those undertaking tree protection measures against the pest ever encountered opposition from the public in this regard?
It appears that the completion of the above information in the submitted work is necessary to justify its advisability. The work is suitable for publication but after appropriate revision.
Author Response
Thank you for bringing attention to our manuscript revisions. We have meticulously incorporated feedback from you, ensuring comprehensive improvements throughout the manuscript. Please refer to the details outlined in the main text file. We are confident that this revised version adequately addresses all queries and enhances the quality of our work.
Question: To what extent does Aeolesthes sarta threaten stands today in the study area? The publications presented on this topic mainly refer to observations made in the 20th century (Kulinich, 1965; Krivosheina, 1984, Gaffar and Bhat, 1991, Orlinski et al. 1991) and early 21st century (Farashiani et al. 2001, EPPO, 2005).
Answer: Thank you for highlighting it. The threat of Aeolesthes sarta standing in the study area persists despite most references dating back to the 20th century and early 21st century. Historical observations, though dated, indicate a potential ongoing risk. Furthermore, the lack of recent publications does not necessarily imply a diminished threat; rather, it may reflect a gap in contemporary research focus or publication trends. Therefore, continued vigilance and research efforts are essential to assess the current extent of the threat posed by Aeolesthes sarta, to stand in the study area, and to implement appropriate management strategies.
To address the current extent of Aeolesthes sarta's threat, we conducted comprehensive field inventories and surveys to gather real-time data on damage rates in the study areas. Although the analysis of this data is ongoing, it will provide valuable insights into the present impact of Aeolesthes sarta. Upon completing our analysis, we intend to publish our findings to contribute to the existing literature on forest pest management. This approach ensures that our assessment of Aeolesthes sarta's threat is based on contemporary evidence, enabling us to formulate informed pest control and forest conservation strategies. By combining historical perspectives with current field data, we aim to comprehensively understand Aeolesthes sarta's threat and facilitate proactive measures to mitigate its impact on forest ecosystems. After the successful publication of this work, we will submit that paper for publication.
Question: Were and what methods of protecting trees from Sart longhorned beetle A. sarta carried out in the study area?
Answer: In QU city, synthetic chemical use for SLB protection prevails on privately owned land, while state-owned forests primarily rely on silvicultural treatments. This disparity underscores the absence of training initiatives for natural forest protection. We have expanded on this observation in both the methodology and discussion sections, highlighting the need for enhanced training and management strategies in state-owned forests. By elucidating these differences, we aim to promote a holistic approach to SLB management, encompassing private and public land management practices to combat SLB damage effectively and safeguard forest health.
Question: Have the green space management services and those undertaking tree protection measures against the pest ever encountered opposition from the public in this regard?
Answer: There was no literature source found that highlighted the encountering of public opposition regarding tree protection measures against the pest in both cities; however, we strongly recommend that there is a need to implement comprehensive educational campaigns tailored to the QU community, emphasizing the risks associated with synthetic pesticides and highlighting the benefits of biological alternatives.
Round 2
Reviewer 2 Report
Comments and Suggestions for Authors
The authors have carefully addressed the recommendations of form and substance in order to improve the quality of the work. I believe that in its current state it qualifies for publication.